# Influence of Environmental Variables on Biochemical Biomarkers in the Amphipod *Monoporeia affinis* from the Gulf of Riga (Baltic Sea)

Evita Strode [1],*, Ieva Barda [1], Natalija Suhareva [1], Natalja Kolesova [2], Raisa Turja [3] and Kari K. Lehtonen [3]

[1] Latvian Institute of Aquatic Ecology, Agency of Daugavpils University, Voleru Str. 4, LV-1007 Riga, Latvia
[2] Department of Marine Systems, Tallinn University of Technology, Akadeemia Tee 15a, 12618 Tallinn, Estonia
[3] Marine Research Centre, Finnish Environment Institute SYKE, Latokartanonkaari 11, FI-00790 Helsinki, Finland
* Correspondence: evita.strode@lhei.lv

**Abstract:** The complexity of the marine environment and the increasing anthropogenic pressure create a necessity to expand existing monitoring approaches. The main goal of this study was to depict the effects of selected, seasonally varying environmental factors on a battery of biomarkers in the benthic amphipod *Monoporeia affinis* from the Gulf of Riga (GoR). Seasonal variability in acetylcholinesterase (AChE), catalase (CAT), glutathione reductase (GR), and glutathione *S*-transferase (GST) activities was investigated at six coastal stations (20–30 m) in August and November in 2020 and 2021. In addition, the biomarkers were measured at seven deep-water stations (>30 m) in November 2021. In general, the results indicated no significant influence of the measured environmental variables on the biomarker activities, except for deep-water stations, where chlorophyll *a* significantly affected enzymatic activity. The current study indicated that *M. affinis* has a higher GST, CAT and GR activity in summer compared to autumn in coastal stations, showing seasonal variability of these biomarkers. However, summarizing the biomarker levels recorded at each station and season, the integrated biomarker response (IBR) index showed the most stressed health status of the *M. affinis* populations in the deep-water stations 135 and 107 and coastal regions in the north-eastern part of the GoR (station 160B). This suggests that the impact on enzymatic responses of benthic organisms could be due to port activities leading to the accumulation of pollutants in muddy sediments regionally. Moreover, for the monitoring of biological effects of contaminants there is a need to establish the background levels of biomarkers, i.e., responses to the different natural environmental factors in the GoR region.

**Keywords:** amphipods; *Monoporeia affinis*; Gulf of Riga; biochemical biomarkers

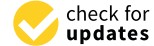



## 1. Introduction

Environmental quality control of aquatic ecosystems receives more and more attention due to increasing economic development and rising pollution. Anthropogenic activities are the main factors in increasing the level of contaminants in aquatic environments [1]. As a result, long-term exposure of aquatic organisms to contaminants leads to changes in physiological responses, accumulation of pollutants in tissues, changes in population level, and, finally, changes in species diversity [2]. Applying molecular, biochemical and physiological biomarkers in environmental monitoring programs helps to detect and identify the sub-lethal early-warning effects of pollution on organisms concerning their health, fitness, growth, and reproductive capacity related to ecosystem health in general [3].

Biochemical biomarkers offer information regarding the potential impact of toxic pollutants on the health of organisms and they represent different types of biological responses to different stressors [4]. A large number of chemical contaminants undergo oxidative reactions in cells, leading to the excess production of reactive oxygen species (ROS), which may cause a common phenomenon called oxidative stress [5]. Antioxidant enzyme activity

is an important part of the antioxidant defense system (ADS) targeted to prevent oxidative damage to cellular components. In ecotoxicology, the most commonly used enzymes to detect oxidative stress include catalase (CAT), glutathione reductase (GR) as well as the phase II biotransformation enzyme glutathione *S*-transferase (GST) [6,7]. The inhibition of acetylcholinesterase activity (AChE), a key enzyme in the cholinergic nervous system, is a classic biomarker of exposure to neurotoxic compounds, particularly to organophosphates and carbamate pesticides [8], but has been recorded to occur also in connection to exposure to other pollutants such as trace metals, detergents and cyanobacterial toxins [9,10]. These common biomarkers are often used to detect exposure to pollution in aquatic organisms in laboratory and field studies [11].

The selection of suitable organisms for a biomonitoring program is an essential step. Amphipods are considered to be excellent bioindicator organisms as they are widespread over large salinity and habitat ranges [12] and respond to various types of environmental contaminants [6]. They are often an important food source for many fish and invertebrate species, and are thus considered highly relevant organisms for ecotoxicological studies [13]. The amphipod *Monoporeia affinis* (Lindström 1855) is an ecological keystone species of the soft-bottom macrozoobenthic communities in the Baltic Sea and thus a relevant bioindicator organism to be used to monitor and assess anthropogenic impacts [14–17]. Malformed embryos in amphipods are currently recommended as a supplementary indicator of contamination effects on aquatic organisms in the Baltic Sea and are regularly monitored by some countries [13,18,19].

For the correct interpretation of biological responses to chemical contamination, it is necessary to recognize the variability in the applied parameters under different natural environmental conditions such as temperature, dissolved oxygen level, salinity, photoperiod and food availability [20–22]. Variability in some responses is also a natural feature of the annual physiological cycle of the species and can be caused by intrinsic confounding factors such as reproductive status [21,23,24]. Regarding *M. affinis*, the combined effects of pollution in the sediment and oxygen deficiency on ADS responses have been recorded using biochemical biomarkers and reproductive disorders both in laboratory experiments [14] and in field studies [18]. In the Gulf of Riga (GoR), the Baltic Sea subregion of the current study, previous studies regarding the effects of pollution on the growth, reproduction and survival of amphipods have been carried out under standardized laboratory conditions [25–28]. However, biochemical biomarker studies from field-collected samples in the area have been carried out only on the soft-bottom clam *Macoma balthica* (Linnaeus, 1758) and the mussel *Mytilus* spp. [10,20,22,26]. Information on the biological effects of contaminants on amphipods from field studies in the Baltic Sea is relatively scarce and, partly therefore, the application of biological effects methods in monitoring in the region has been underdeveloped until very recently. The research on biochemical biomarkers in *M. affinis* would be the first information in the GoR region and could therefore be used as the basis of regular biomonitoring of contamination effects in the area for future.

The aim of this study was to assess seasonal and spatial variability in selected biochemical biomarkers in *M. affinis* in the GoR as a possible starting point for regular monitoring activities in the area. The above-mentioned ADS response biomarkers CAT, GR and GST were selected to detect possible contaminant-induced oxidative stress while AChE inhibition represents exposure to directly neurotoxic substances or indirect inhibitory effects caused by other types of compounds. The samplings were carried out in 2020 and 2021 in summer and autumn in both years, using a network of sampling sites consisting of shallow coastal and deep-water areas of the GoR. A common set of environmental variables were recorded simultaneously to examine the relationships of these factors with the variability observed in the biomarker levels in the study area.

## 2. Materials and Methods

### 2.1. Study Area

The Gulf of Riga is located in the northeastern part of the Baltic Sea and is a semi-enclosed basin with a relatively low salinity of 0.5–7.7 ppt due to its isolation from the Baltic Proper and significant freshwater discharges [29,30]. The GoR surface area is 16.330 km$^2$ (3.9% of the Baltic Sea area), water volume 424 km$^3$ (2.1% of the Baltic Sea volume), average depth 26 m, and maximum depth 62 m [31]. Near-bottom water temperature remains low all year round (4–6 °C) [32,33] and dissolved oxygen concentrations range from 2 to 6 mg/L [31,34]. At the depths of 20–30 m, soft bottom substrates or sandy sediments and boulders with well-developed benthic communities prevail with richness up to 13 species, while silty and muddy sediments mostly dominate in the deeper parts (>30 m) of the gulf consisting of only a few amphipod and polychaete species [27,32,35,36].

### 2.2. Sampling

The *M. affinis* sampling stations were chosen based on the species abundance data available from the Latvian national marine monitoring. The amphipods were collected aboard the R/V "Salme" in August and November in 2020 and 2021 with a Van Veen grab (0.1 m$^2$) at the depths of 20–43 m (Figure 1) and sieved through a 0.5 mm mesh. The amphipods were immediately frozen in liquid nitrogen, and later placed in a −80 °C freezer for the later biomarker analyses. Salinity, temperature, chlorophyll *a*, and dissolved oxygen concentration, in the near-bottom layer were measured at each site with a multiparameter water quality probe CTD profiler SBE 19 plus SeaCAT (Bellevue, WA, USA).

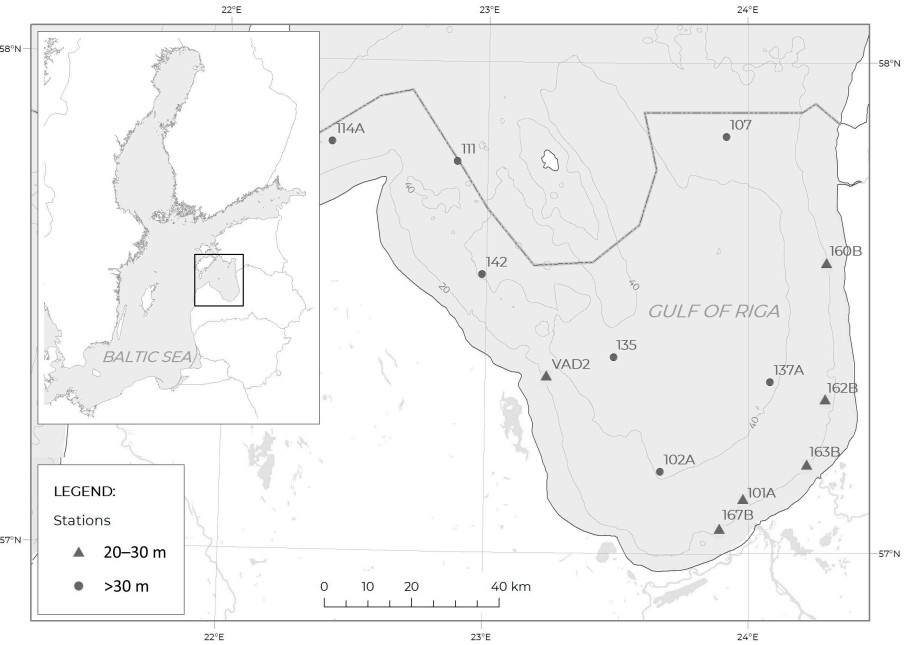

**Figure 1.** Map of the study area in the Gulf of Riga with markings of coastal (20–30 m) and deep-water (>30 m) stations.

### 2.3. Sample Preparation and Biomarker Analyses

Whole bodies of five amphipods were pooled for each of the 12 replicate biomarker samples per station. Six replicates were homogenized for 2 × 45 s (Retsch MM400 homogenizer, Haan, Germany) in cold (4 °C) 0.1 M phosphate buffer with 0.15 M KCl (pH 7.4) (1:4 *w/v* ratio) to measure the ADS enzyme activities. The remaining six replicates were used to determine AChE with homogenization using cold 0.02 M phosphate buffer with 0.1% Triton X-100 (pH 7.0). During homogenization, the vials were kept on ice. The homogenate was centrifuged at 12,000× *g* at 4 °C for 20 min and the supernatant was used for the assays. Four measurements were carried out per replicate. The enzyme activities

were measured with a microplate reader (Spark® Multimode Microplate Reader, TECAN, Grödig, Austria) and analyzed using Magellan software (software version 2.2). The reaction rate was evaluated according to the best linear range of the obtained curve. The specific activities of all the enzymatic biomarkers were calculated against the total protein content of the sample.

The CAT activity was determined according to Claiborne [37]. The supernatant was diluted with the phosphate buffer in a 1:10 (*v/v*) ratio. The CAT activity was measured recording the decrease of 30 mM $H_2O_2$ at 240 nm. The measurement of GST activity is based on the conjugation of reduced glutathione (GSH) to 1-chloro-2,4-dinitrobenzene (CDNB) using a modification of the method based on Habig et al. [38]. The supernatant in a reaction mix (Dulbecco's buffer PBS, 0.1 M GSH and 0.1 M CDNB) were used to measure GST activity at 340 nm. The GR activity level was evaluated as the oxidation rate of NADPH. The supernatant in a reaction mix (0.1 M phosphate buffer + 2 mM EDTA [pH 7.5], 2 mM GSSG, 2 mM NADPH, 3 mM DTNB) was analyzed for GR activity at 412 nm [39]. For the determination of AChE activity according to Ellman et al. [40], the supernatant in 0.02 M phosphate buffer (pH 7.0), 0.1 M acetylcholine iodide (ACTC) and 0.01 M 5,5'-dithiobis (2-nitrobenzoic acid) (DTNB) was measured at a 412 nm.

The total amount of protein in the homogenate of each sample was determined with the Bradford assay using bovine serum albumin as the standard [41].

### 2.4. Statistical Analysis

The biomarker levels were expressed as the mean and standard error (mean ± SE) recorded for amphipods collected from each station by each month, year, and the coastal/ deep-water grouping. To ensure homogeneity of data while assessing the seasonal and interannual variability based on the grouped data of the biomarker activity of all coastal stations together, only stations 101A, 167B, VAD2, 163B, 162B were considered in the calculations. The environmental variables recorded were expressed as the mean and standard deviation (mean ± SD). The normality of the biomarker data distribution was checked via the Shapiro–Wilk test. As most of the data were not normally distributed, non-parametric tests were applied. Due to failed data normality, the non-parametric Kruskal–Wallis one-way analysis of variance in combination with the Wilcoxon rank sum test was performed to investigate seasonal and annual variability in medians of biomarker activities across the entire GoR and at the individual sampling stations. Correlations between biomarker levels and physicochemical variables were examined by means of the Spearman's rank correlation and visualized as a correlation matrix using the "corrplot" package of the R software. Data exploration, artworks, and statistical analyses were performed using the R software for Windows, release 4.0.3.

All the measured biomarkers were combined into one general "stress index" known as the Integrated Biomarker Response (IBR) index [42]. The procedure used for the IBR calculation was based on the original paper by Beliaeff and Burgeot [42], modified by Broeg and Lehtonen [43]. The AChE, GST, CAT, and GR data from the two seasons (August and November) in the year 2020 were used for the IBR calculation for the coastal stations while the November data from the year 2021 were used to compare IBR at the coastal and deep-water stations.

## 3. Results

### 3.1. Environmental Factors

The measured physicochemical parameters (Table 1) varied moderately between the two seasons (August and November) and the station depth group. The data presented in Table 1 shows that both compared years, 2020 and 2021, differed significantly in terms of water temperature: in August the median in the group of coastal water sites (20–30 m depth) was respectively 8.1 °C and 3.3 °C, while in November it was 10.8 °C and 8.4 °C, respectively. At the coastal stations, higher average values in temperature, chlorophyll *a* and oxygen level were observed in November (9.6 ± 0.4 °C, 2.6 ± 0.2 mg/m³, and 10.4 ± 0.2 mg/L,

respectively) compared to those measured in August (7.5 ± 1.4 °C, 1.3 ± 0.3 mg/m$^3$ and 5.5 ± 0.5 mg/L, respectively). In addition, in 2021, higher concentrations of dissolved oxygen and chlorophyll *a* in water were recorded in all coastal stations compared to the year 2020. A similar trend was recorded between the depth groups in November, with higher values in temperature, chlorophyll *a* and oxygen level being detected at the coastal stations compared to the deep-water stations (8.1 ± 0.5 °C, 2.2 ± 0.2 mg/m$^3$, and 9.1 ± 1.2 mg/L, respectively). The average water temperature was slightly lower (by 0.4 °C) in deep-water sites in comparison to coastal sites. At the same time, the median in both groups in (November 2021) of stations (coastal and deep-water) was identical (8.4 °C). However, in November 2021 the higher concentrations of chlorophyll *a* and oxygen content were recorded at coastal stations.

**Table 1.** Physicochemical parameters measured at the different study stations in the near-bottom layer in August and November 2020 and 2021.

| Year | Month | Station | Depth [m] | Temp. [°C] | Absolute Salinity [ppt] | Chlorophyll *a* [mg/m$^3$] | Oxygen [mg/L] |
|---|---|---|---|---|---|---|---|
| 2020 | August | 101A | 22 | 7.2 | 5.8 | 0.9 | 4.0 |
| 2021 | | | 22 | 3.3 | 6.1 | 1.8 | 5.9 |
| 2020 | November | | 22 | 10.9 | 6.0 | 1.6 | 9.5 |
| 2021 | | | 23 | 8.4 | 6.0 | 3.7 | 11.1 |
| 2020 | August | 167B | 21 | 6.4 | 5.8 | 0.8 | 4.7 |
| 2021 | | | 21 | 3.2 | 6.2 | 1.6 | 6.4 |
| 2020 | November | | 21 | 10.8 | 5.9 | 2.0 | 9.7 |
| 2021 | | | 22 | 8.4 | 6.0 | 3.0 | 11.1 |
| 2020 | August | 163B | 22 | 12.1 | 5.9 | 0.9 | 6.4 |
| 2021 | | | 22 | 18.4 | 5.9 | 2.2 | 6.9 |
| 2020 | November | | 22 | 10.8 | 5.9 | 2.1 | 9.7 |
| 2021 | | | 21 | 8.5 | 5.9 | 3.0 | 11.1 |
| 2020 | August | 162B | 25 | 8.9 | 6.0 | 0.8 | 4.0 |
| 2021 | | | 25 | 3.9 | 6.1 | 1.5 | 6.8 |
| 2020 | November | | 25 | 10.7 | 5.9 | 1.9 | 9.6 |
| 2021 | | | 24 | 8.3 | 5.9 | 4.0 | 11.3 |
| 2020 | August | VAD2 | 26 | 5.5 | 5.9 | 0.9 | 4.6 |
| 2021 | | | 26 | 3.0 | 6.2 | 1.5 | 8.3 |
| 2020 | November | | 26 | 10.3 | 5.9 | 2.3 | 9.8 |
| 2021 | | | 25 | 7.7 | 6.1 | 3.0 | 11.1 |
| 2020 | August | 160B | 22 | 10.1 | 5.9 | 0.9 | 2.9 |
| 2020 | November | | 22 | 10.4 | 5.7 | 2.4 | 9.9 |
| 2021 | November | | 22 | 7.9 | 5.9 | 3.0 | 11.5 |
| 2021 | November | 107 | 32 | 8.4 | 6.0 | 2.7 | 11.2 |
| 2021 | November | 111 | 38 | 8.8 | 6.5 | 2.0 | 10.7 |
| 2021 | November | 114A | 33 | 8.6 | 6.8 | 1.7 | 10.9 |
| 2021 | November | 142 | 42 | 8.2 | 6.3 | 2.3 | 10.7 |
| 2021 | November | 135 | 45 | 8.2 | 6.1 | 2.6 | 5.3 |
| 2021 | November | 102A | 42 | 5.3 | 6.2 | 1.6 | 4.0 |
| 2021 | November | 137A | 42 | 9.0 | 6.0 | 2.2 | 10.7 |

### 3.2. Seasonal Variability in Biomarkers at the Coastal Stations

In *M. affinis*, the median AChE activity of the pooled data of coastal stations showed no significant variability between August and November ($p > 0.05$) in both year (Figure 2). Although, looking at each station separately, the AChE activity level at station 160B was lower in August than in November in 2020 ($p = 0.020$) and at station 163B in 2021 ($p = 0.01$). Interannual variability could be seen both in August ($227.5 \pm 7.6$ and $191.9 \pm 8.3$ nmol/min/mg/protein, in 2020 and 2021, respectively, $p = 0.005$) and November ($242.1 \pm 11.2$ and $179.3 \pm 7.4$ nmol/min/mg/protein, $p < 0.001$).

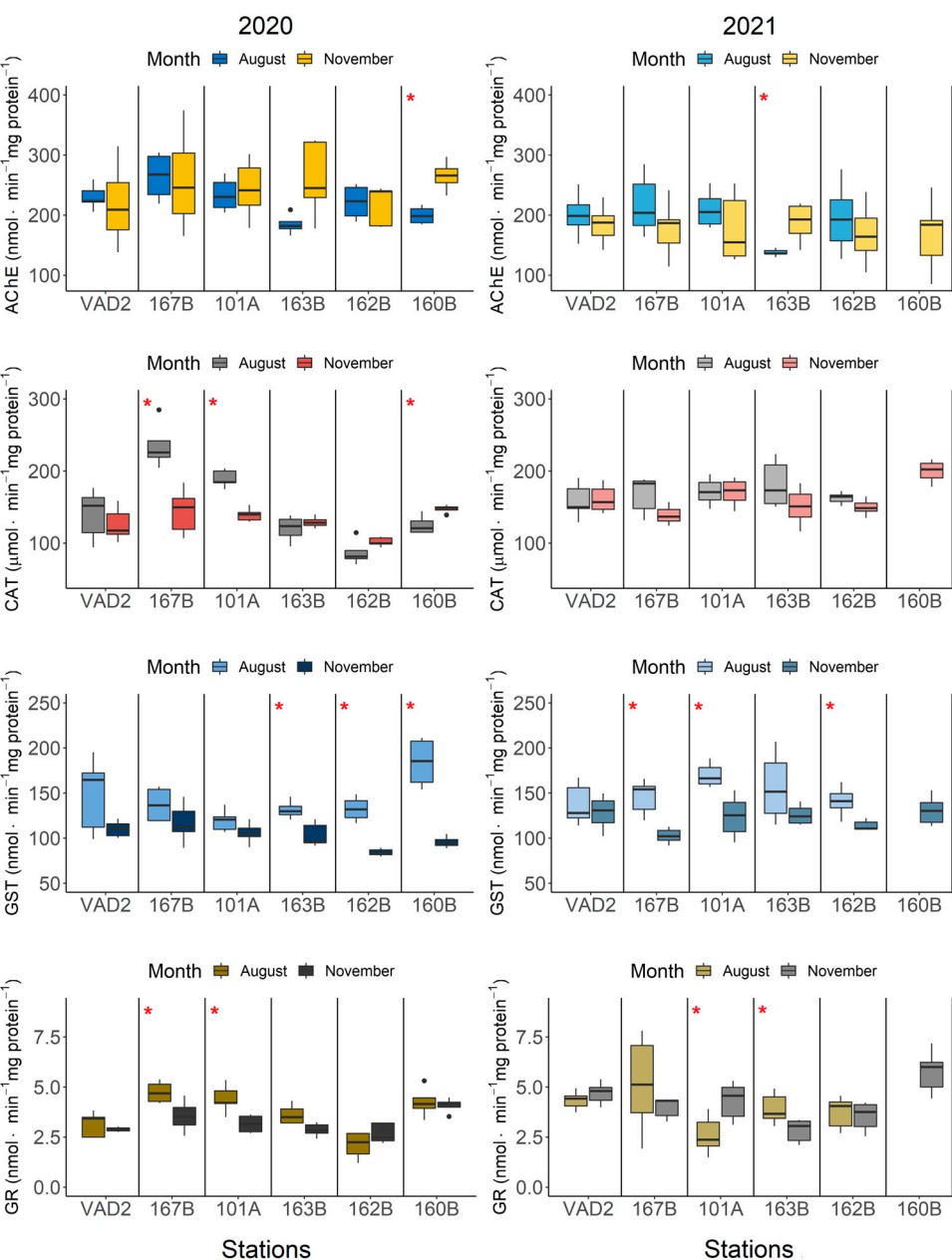

**Figure 2.** Activity of AChE, CAT, GST, and GR in *M. affinis* at the coastal stations (20–30 m) in the GoR in August and November 2020 and 2021. The red asterisk indicates statistically significant ($p < 0.05$) seasonal differences at the respective stations. Black dots denote outliers calculated according to the Interquartile range (IQR) criterion.

Grouped coastal stations CAT activity did not show any significant variability between August and November in 2020, however it was significant ($p = 0.017$) in 2021 (Figure 2). In 2020, a higher activity was recorded in August at some individual stations,

namely 101A ($p = 0.009$), 160B ($p = 0.043$), and 167B ($p = 0.008$). No significant differences between the years could be found in the August samples but in November 2020 the activity was significantly lower ($p < 0.001$) compared to November 2021 ($126.9 \pm 3.9$ and $154.4 \pm 3.6$ µmol/min/mg/protein, respectively).

In GST, significant interannual (2020 versus 2021) variability has been observed for both August ($p = 0.019$) and November ($p = 0.003$) (Figure 2). Seasonal variability was significant ($p < 0.001$) both for 2020 ($133.9 \pm 5.1$ and $105.1 \pm 3.2$ nmol/min/mg/protein in August and November, respectively) and 2021 ($148.8 \pm 3.9$ and $123.8 \pm 2.4$ nmol/min/mg/protein). Seasonal differences in GST activity were significant at most of the coastal stations including 160B ($p = 0.021$), 162B ($p = 0.014$), and 163B ($p = 0.027$) in 2020, and 101A ($p = 0.004$), 162B ($p = 0.010$), 167B ($p = 0.004$) in 2021.

*Monoporeia affinis* at the individual stations 101A and 167B in 2020 ($p = 0.028$ and 0.037, respectively) and 101A and 163B in 2021 ($p = 0.024$ and 0.016) showed higher GR activity in August (Figure 2). Interannual variability was significant in the November samples where the GR activity across the stations was higher in 2021 ($3.9 \pm 0.2$ nmol/min/mg/protein) compared to 2020 ($3.3 \pm 0.1$ nmol/min/mg/protein) ($p < 0.001$).

### 3.3. Biochemical Biomarkers at the Deep-Water Stations

Biomarker levels in *M. affinis* at the deep-water stations were measured only in November 2021 (Figure 3). The highest values of AChE activity were observed at station 135 ($262.2 \pm 24.6$ nmol/min/mg/protein), being significantly higher ($p < 0.05$) than at stations 114A, 137A and 111 ($115.0 \pm 7.8$, $140.3 \pm 7.1$ and $163.9 \pm 12.4$ nmol/min/mg/protein, respectively).

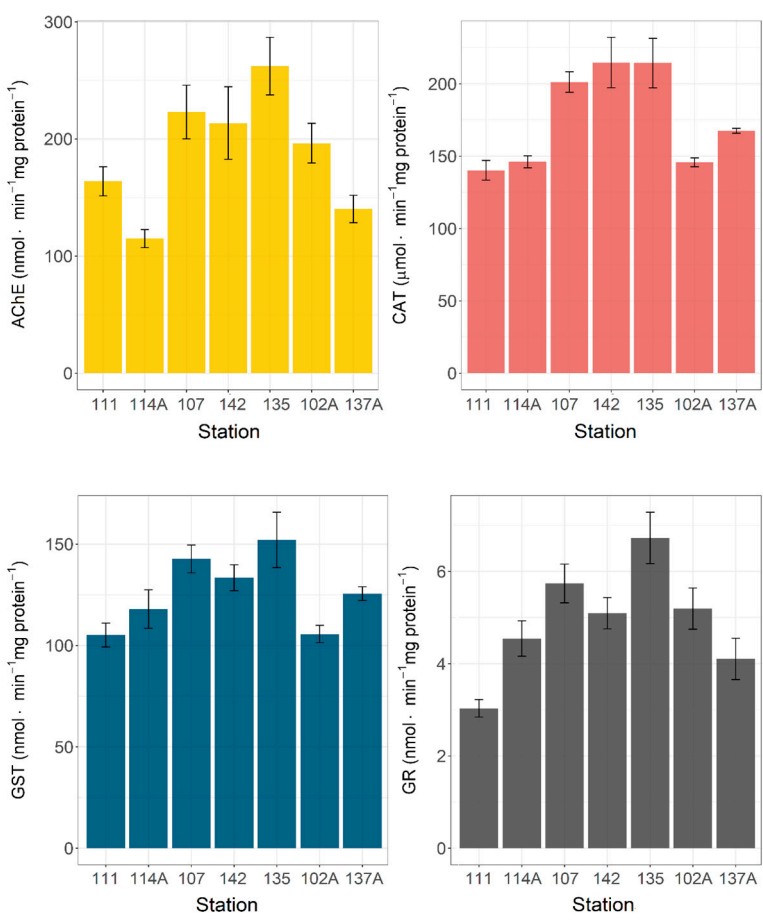

**Figure 3.** Activity (mean $\pm$ SE) of AChE, CAT, GST, and GR in *M. affinis* at the deep-water stations in November 2021.

For CAT, the activity levels were higher at stations 107, 135 and 142 (from $201.3 \pm 7.1$ to $214.7 \pm 17.2$ µmol/min/mg/protein) compared to the other four stations ($p < 0.01$).

The GST activity was significantly higher ($p < 0.05$) in samples collected from stations 107 and 135 ($142.7 \pm 6.9$ and $152.1 \pm 13.7$ nmol/min/mg/protein, respectively), compared to stations 102A and 111 ($105.7 \pm 4.30$ and $105.2 \pm 5.9$ nmol/min/mg/protein) (Figure 3).

The GR activity was significantly lower ($p = 0.011$ to $0.031$) in samples collected from station 111 ($3.0 \pm 0.2$ nmol/min/mg/protein) compared to the other stations ($5.2 \pm 0.4$, $5.7 \pm 0.4$, $4.6 \pm 0.4$, $6.7 \pm 0.6$, $5.1 \pm 0.3$ nmol/min/mg/protein for stations 102A, 107, 114A, 135 and 142, respectively), except 137A ($4.1 \pm 0.5$ nmol/min/mg/protein).

Moreover, the activity of CAT and GR among the deep-water stations was significantly higher ($p < 0.001$) than it was among the coastal stations in November 2021.

### 3.4. Correlation Analysis

Only some of the physicochemical variables showed strong or moderate correlations within a season at the coastal stations (Figure 4). In general, more correlations were found in November compared to August. Near-bottom oxygen concentrations showed a negative correlation with AChE activity of the amphipods both in August and November while being positively correlated with CAT and GR activity in November. Moreover, chlorophyll *a* concentration exhibited a strong negative correlation with AChE and was positive with CAT in November, while no significant relationships between the variables could be observed in August. Regarding temperature, only AChE activity in November showed a significant positive correlation. In the deeper areas of the gulf in November, chlorophyll *a* concentration showed a significant positive correlation with AChE, CAT, and GST (Figure 4).

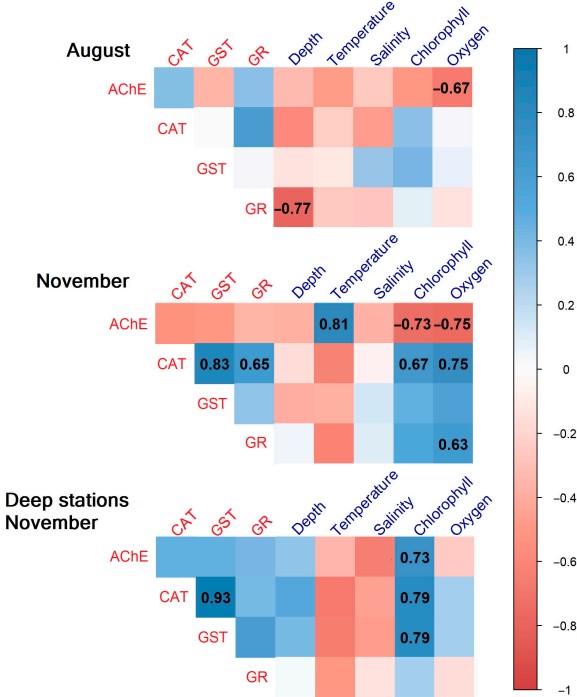

**Figure 4.** Spearman's rank correlation coefficients between biomarker levels and physicochemical variables were calculated for six coastal stations in August and November, and for seven deep-water stations in November. Correlation coefficients (Spearman's rho) with a $p < 0.05$ are shown and marked based on their sign (+ or −) and strength (continuous color scale).

### 3.5. Integrated Biomarker Response

The IBR calculations showed variability in the integrated response at the different stations, seasons, and depth ranges (Figure 5). Across the coastal stations, higher index

values were detected in August at stations 160B and 167B (Figure 5a). In November, the most impacted amphipod population was observed at the same station 160B while the least impacted was 162B in both seasons. In the comparison of coastal and deep-water station data in November, higher IBR index values were commonly observed at the deep-water stations, especially at 135, 107 and 142, with the notable exception of station 111 (Figure 5b).

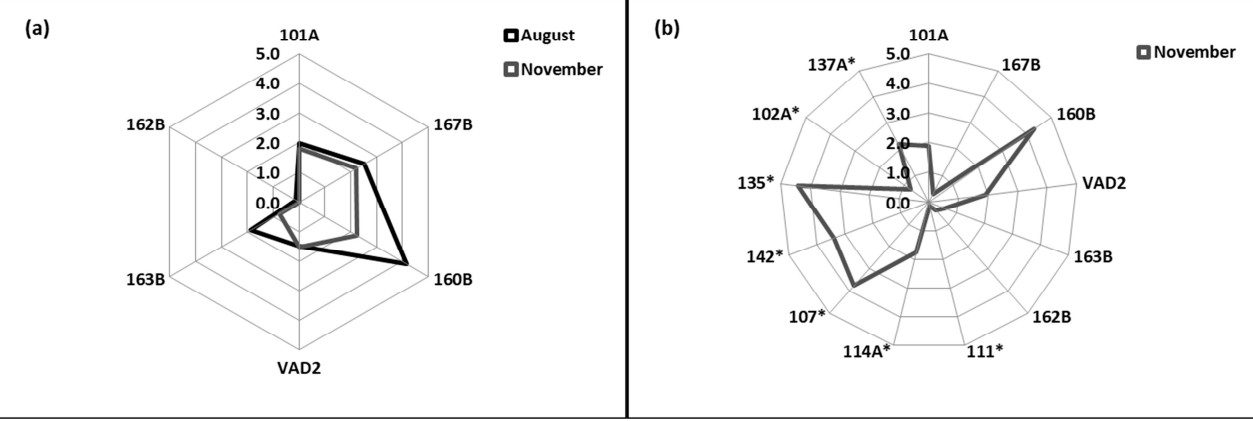

**Figure 5.** Integrated biomarker response index (IBR) in the amphipod *M. affinis* collected from (**a**) coastal, year 2020 and (**b**) coastal and deep-water stations (marked *), year 2021 from the GoR.

## 4. Discussion

This study focused on the spatial and temporal differences in selected biomarker responses in a key species in the Gulf of Riga study area and the associations of the responses with some environmental factors. The results obtained provide the first information on biochemical biomarkers in *M. affinis* in this sea area and can therefore be used as the basis of regular biomonitoring of contamination effects in the area.

Seasonal variability has been recognized as an important factor influencing the baseline levels of biomarkers and also the responsiveness of organisms to pollution stress [21,44,45], e.g., in previous studies on *M. balthica* from the GoR [25]. In the present study, seasonal variability in GST and CAT was detected in the amphipods at all the study stations. Of the effects of specific environmental variables measured, the impact of chlorophyll *a* on AChE, CAT, and GST was observed in the samples collected in November in the deeper parts of the GoR. At the same time, a significant negative correlation between chlorophyll *a* in the bottom layer and AChE activity of the amphipods was detected at the coastal stations in November. Normally, a high chlorophyll *a* level in water would indicate good feeding opportunities for filter and deposit feeders and result in an improved physiological condition; however, if associated with toxic phytoplankton blooms such as those regularly occurring in the Baltic Sea in late summer it can cause the inhibition of AChE due to residual cyanobacterial toxins [9,10,14]. Among other abiotic factors, Löf et al. [46] found that both salinity and temperature significantly modified some biomarker responses to contaminants in *M. affinis*. Negative effects of increased temperature during contaminant exposure on *M. affinis*, both on a biochemical and organism level, were recorded also in laboratory conditions [47]. However, in our study, temperature positively and oxygen negatively correlated with AChE activity in November at the coastal stations. A rise in temperature can increase the oxygen consumption of the amphipods, thus increasing energy cost and inhibiting quality of reproductivity [48].

Apart from abiotic variables, biotic factors such as reproductive status, body size and food availability can affect enzymatic responses in amphipods [14,49–51]. Therefore, the physiological status of the organisms should be taken into account in the interpretation of biomarker responses in field studies [52]. *Monoporeia affinis* has a long reproductive cycle; oogenesis starts in late summer and mating takes place in November [53], and the physiological processes during the breeding period lead to the mobilization of energy

stores within the organism, potentially increasing their sensitivity to environmental stressors [52]. Significant seasonal differences were also detected in the activity of enzymatic biomarkers between summer and autumn in the clam *M. balthica* from the GoR [20] and the Gulf of Finland [22]. The current study indicated that *M. affinis* has a higher GST, CAT and GR activity in summer compared to autumn. For the amphipod *Hyalella kaingang*, Braghirolli et al. [21] reported a significantly higher CAT activity in summer compared to autumn; however, opposite results were obtained regarding GST activity. Conclusively, the observations above emphasize the importance of being aware of possible different seasonal baselines in biomarkers that can depend on various abiotic and biotic factors [54].

Various biological processes provide information on the types of stressors affecting benthic soft-bottom communities and the reproduction of amphipods possibly related to their observed population decreases in the Baltic Sea [17,55]. Parallel studies in the GoR (unpublished data) indicate that the reproductive success of *M. affinis* is moderately associated with the measured biomarker levels under natural conditions and this information can be utilized when designing early warning monitoring schemes using these parameters. Although the observations on the variability in biomarkers presented in this study are not temporally broad enough to draw conclusions on their long-term dynamics, the significant differences observed in all four biomarker responses between 2020 and 2021 highlight the necessity for longer-term monitoring to reveal the drivers of the observed population declines.

The central part of the GoR (i.e., the deep-water stations) is characterized by a low-diversity benthic community, muddy sediments and relatively high trace metal concentrations [27], where pollutants are relocated and accumulated by currents. The integrated stress response (IBR) in amphipods, assessed by combining the information obtained from all four biomarkers selected for this study, shows a peak in autumn and is more pronounced in the deep-water areas. The elevated biomarker IBR responses were generally observed in *M. affinis* collected from the central region of the GoR (deep-water stations 135, 142) and the north-eastern part of the GoR (stations 160B and 107) as well as at stations situated in the river estuaries. In many cases the biomarker levels were higher at the deep-water stations compared to the coastal area. Elevated trace metal concentrations, particularly of Cd, have been recorded in sediments at the deep-water areas as well as at one sublittoral station (160B) [27]. Obviously, there are also many other contaminants present in the GoR marine environment. Using trace metal levels in sediments as a rough proxy of anthropogenic chemical pollution and their apparent association with the biomarker responses, including the IBR, gives an indication of the usefulness of the biomarker approach when assessing the biological effects of environmental contamination in the area. However, as shown in this study, natural variability has also to be carefully considered when interpreting the results.

## 5. Conclusions

The current study represents the first comprehensive report on a battery of biochemical biomarkers measured in *M. affinis* collected from the GoR at different seasons and years. The results showed that biological effects methods can be used in environmental quality assessment in this Baltic Sea area. Higher response levels were recorded in populations inhabiting the central (deeper) and northeastern regions of the GoR. Furthermore, fewer differences in biomarker levels were observed in amphipods collected in August compared to November, indicating apparent seasonal variability. The observations highlight the importance of understanding the physiology of a species in its natural habitat, including the periods of greater vulnerability to environmental stressors due to biotic and abiotic factors. Using a specifically designed battery of biomarkers together with measuring a variety of contaminants and key physicochemical variables in monitoring improves our understanding of biochemical responses to environmental stress. This research shows that the use of amphipods for integrated chemical-biological monitoring as well as for the evaluation of sediment quality of the GoR is possible. Finally, the results contribute to the knowledge on the effects of environmental factors on biomarker responses in amphipods

and to clarify their dependency on the season that will lead to the development of their natural baseline/threshold values needed in biomonitoring.

**Author Contributions:** E.S.: conceptualization, data curation, resources, project administration, funding acquisition, formal analysis, validation, visualization, writing—original draft, writing—review and editing; I.B.: conceptualization, resources, validation, visualization, writing—original draft, writing—review and editing; N.S.: data curation, software, visualization, writing—original draft, writing—review and editing. N.K.: validation, writing—review and editing; R.T.: methodology, writing—review and editing; K.K.L.: investigation, methodology, writing—review and editing. All authors have read and agreed to the published version of the manuscript.

**Funding:** This research was funded by the European Regional Development Fund, 1.1.1.2/16/l/001 Post-doctoral project No. 1.1.1.2/VIAA/3/19/465.

**Data Availability Statement:** The data presented in this study are available on request from the corresponding author. The data are not publicly available due to privacy or ethical restrictions.

**Acknowledgments:** Our special thanks to the Latvian Institute of Aquatic Ecology laboratory assistants and researchers who participated in the sediment and amphipods sampling, as well as colleagues who made physicochemical analyses.

**Conflicts of Interest:** The authors declare no conflict of interest.

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
