# Peer review of "Influence of Environmental Variables on Biochemical Biomarkers in the Amphipod Monoporeia affinis from the Gulf of Riga (Baltic Sea)"

_water, doi:10.3390/w15020248_

Round 1
Reviewer 1 Report
The manuscript presents the results of research a battery of biomarkers in the benthic amphipod Monoporeia affinis from the Gulf of Riga (GoR) in 2020-2021. Their goal was to determine the natural background conditions and to indicate whether this set of markers can be used in the assessment of the environmental quality of this bay.
I found the manuscript interesting, newsworthy. However, I suggest considering a few tweaks:
Lines 157-162: I'm guessing these rules didn't apply to 160B, for which data was only collected in 2020. But don't such comparisons distort the obtained results? Is an identical picture obtained for data from 2020 only? As the graphs in Figure 3 show, there is a variation in biomarker values between both years.
Lines 170-173: The data presented in Table 1 shows that both compared years 2020 and 2021 differed significantly in terms of water temperature: in August the median (due to the huge standard error of the average) in the group of shallow sites (20-30 m) was respectively 7.2°C and 3.3°C, while in November - 10.8°C and 8.4°C (by the way, similar to the average with SD = 0.2-0.3°C). In addition, in 2021, higher concentrations of dissolved oxygen and chlorophyll a in water were always recorded.
Therefore, the comparison of deep and shallower sites can only be made on the basis of data from 2021. In this situation, the average water temperature is slightly lower (by 0.3°C) in the group of deep sites, however, it results only from one outlier measurement (5°C) at point 102A. In two other points, the temperature was higher than in the group of shallower points, and the median in both groups of points is identical (8.4°C). However, in fact, this year higher concentrations of chlorophyll a and oxygen content were recorded at coastal points.
Lines 179-180: The cited figure 2 shows (red star at position 160B) that there was some variation in the grouped AChE activity, although the next sentence shows that it concerned 2020. Similarly for point 163B in 2021. These discrepancies need to be corrected.
Lines 182-184: Were the reported differences between years calculated from the same five stations, or did the 2020 calculations also include data from the 160B station, which was not tested in 2021?
Line 185: I suggest placing the dates (2020 and 20221) above the appropriate columns of the drawings.
Lines 186-187: The caption of the figure should explain what the dark dots above the boxplots mean. By default, these are outliers, but it should be indicated according to what criterion - also standard in R?
Lines 189-190: Maybe I don't understand the authors' intentions, but in 2020, almost 43% of the sites surveyed showed differences between August and November. Is it therefore possible to formulate the first sentence of the paragraph in this way, claiming that there were no differences? In fact, they were absent only in 2021.
Lines 196-197: Based on this sentence and the graph in Figure 2, the following questions arise: If the data pair is marked with a red asterisk in the graph, does it mean a significant difference between August and November? If so, on what basis was it indicated that the differences concerned only November? Was the homogeneity of the August results examined?
Author Response
Dear reviewer,
We thank you for the feedback and constructive criticism obtained for our manuscript. We have taken your suggestions into account and hope that you now find the manuscript suitable for publication in Water.
Listed below are your comments and our responses.
Point 1: Lines 157-162: I'm guessing these rules didn't apply to 160B, for which data was only collected in 2020. But don't such comparisons distort the obtained results? Is an identical picture obtained for data from 2020 only? As the graphs in Figure 3 show, there is a variation in biomarker values between both years.
Response 1: Agreed. We recalculated the IBR index separately between the years and found that results did not change significantly – tendencies are indeed the same. Accordingly, text changes were made in the result’s paragraph. Namely, station 163B was taken out of the text and the caption and data behind Figure 5 A and B were modified.
Point 2: Lines 170-173: The data presented in Table 1 shows that both compared years 2020 and 2021 differed significantly in terms of water temperature: in August the median (due to the huge standard error of the average) in the group of shallow sites (20-30 m) was respectively 7.2°C and 3.3°C, while in November - 10.8°C and 8.4°C (by the way, similar to the average with SD = 0.2-0.3°C). In addition, in 2021, higher concentrations of dissolved oxygen and chlorophyll a in water were always recorded.
Response 2: Agreed. We added one row of physicochemical data from 160B station in November 2021 in Table 1. The corresponding result section was supplemented with information suggested by the reviewer.
Point 3: Therefore, the comparison of deep and shallower sites can only be made on the basis of data from 2021. In this situation, the average water temperature is slightly lower (by 0.3°C) in the group of deep sites, however, it results only from one outlier measurement (5°C) at point 102A. In two other points, the temperature was higher than in the group of shallower points, and the median in both groups of points is identical (8.4°C). However, this year higher concentrations of chlorophyll a and oxygen content were recorded at coastal points.
Response 3: The suggested information provided by the reviewer was incorporated in the rewritten result part, alongside changes made in response to Point 2.
Point 4: Lines 179-180: The cited figure 2 shows (red star at position 160B) that there was some variation in the grouped AChE activity, although the next sentence shows that it concerned 2020. Similarly for point 163B in 2021. These discrepancies need to be corrected.
Response 4: Thank you for highlighting areas for further clarification. The first sentence of the paragraph should say that the overall gulf median (pooled data of all stations) does not differ statistically, whereas the second sentence should say that some significant seasonal variabilities were found when each station was looked at separately. These sentences have been rephrased.
Point 5: Lines 182-184: Were the reported differences between years calculated from the same five stations, or did the 2020 calculations also include data from the 160B station, which was not tested in 2021?
Response 5: Initially calculations included data from station 160B, however, as no data were recorded in August 2021 for this station, results were recalculated without station 160B. An additional sentence explaining removal of station 160B from the calculations was added to the methods section about reported differences calculated between years. Changes in the biomarker values were introduced in the relevant places in the results section (lines 206-209; 219-222; 227-229; 235-237).
Point 6: Line 185: I suggest placing the dates (2020 and 2021) above the appropriate columns of the drawings.
Response 6: We took the reviewers' suggestions into account, that is, years 2020 and 2021 have been placed above the appropriate columns of the drawings, and the data from the station 160B of November 2021 were added.
Point 7: Lines 186-187: The caption of the figure should explain what the dark dots above the boxplots mean. By default, these are outliers, but it should be indicated according to what criterion - also standard in R?
Response 7: Dark dots above the boxplots are outliers and they were calculated according to the Interquartile range (IQR) criterion, standard for the boxplot() function of the ggplot2 package in R. The Figure 2 caption has been changed accordingly.
Point 8: Lines 189-190: Maybe I don't understand the authors' intentions, but in 2020, almost 43% of the sites surveyed showed differences between August and November. Is it therefore possible to formulate the first sentence of the paragraph in this way, claiming that there were no differences? In fact, they were absent only in 2021.
Response 8: Probably, the sentence has a confusing formulation. The first sentence of the paragraph says that the overall gulf median (the pooled data of all stations) does not differ statistically. Meanwhile, in the second sentence it is stated that by looking at each station separately, some significant seasonal variabilities were found. The sentence has been rephrased.
Point 9: Lines 196-197: Based on this sentence and the graph in Figure 2, the following questions arise: If the data pair is marked with a red asterisk in the graph, does it mean a significant difference between August and November? If so, on what basis was it indicated that the differences concerned only November? Was the homogeneity of the August results examined?
Response 9: As it is described in the caption of Figure 2, the red asterisk in the graph indicates statistically significant seasonal variation between August and November in each individual station, in years 2020 and 2021 separately.
Whereas the sentence in lines 196-197 discusses interannual variation between years 2020 and 2021, separately in August and November samples. As it was mentioned in the sentence, the homogeneity of the August samples was examined similarly as the homogeneity of November samples. The sentence as well as the Figure 2 caption have been rephrased after data recalculation without station 160B.

Reviewer 2 Report
1. It is suggested that the author emphasize the novelty of this paper in the introduction.
2. Is the selected study area representative?
3. More important findings and data should be included in the abstract.
4. It is recommended to further change the format of references. Some articles do not have DOI.
5. There are many abbreviations in this article. If possible, it is recommended to add an abbreviation summary.
Author Response
Dear reviewer,
We thank you for the feedback and constructive criticism obtained for our manuscript. We have taken your suggestions into account and hope that you now find the manuscript suitable for publication in Water.
Listed below are your comments and our responses.
Response to Reviewer 2 Comments
Point 1: It is suggested that the author emphasize the novelty of this paper in the introduction.
Response 1: Reviewers’ comment was taken into account and an additional sentence has been added to the introduction to emphasize novelty in lines 91-98.
- Is the selected study area representative?
Response 2: Yes, the study area is representative. The Gulf of Riga is one of the subbasins of Baltic Sea; a relatively shallow bay connected to the deeper central Baltic Sea (Baltic Proper) via straits with sills where seasonal hypoxia occasionally occurs. This bay is typical of environmental conditions with the stratified season and haline stratification in the deep layer. Because of the shallowness of the basin, the whole water column is well mixed and vertical distributions of temperature, salinity and oxygen are homogenous in the winter. In the summer, stratification is mainly maintained by the seasonal thermocline, which starts to develop in April and is the strongest in August, and the contribution of haline stratification is rather moderate. The amphipods M. affinis are the main habitant of coastal and deepwater areas and the sampling sites chosen in this study represent almost entirely the Gulf of Riga.
- More important findings and data should be included in the abstract.
Response 3: Reviewers’ comment was taken into account and significant findings and data have been added to the abstract (lines 19-21; 21-24).
- It is recommended to further change the format of references. Some articles do not have DOI.
Response 4: Agreed. Thank you for your recommendation. We checked the DOI in the reference section. Unfortunately, for some references (Number 17, 29 and 31) it was not possible to find DOI. The DOI numbers were added for reference numbers 12.,18.,19.,32., 35., 38., and 42.
- There are many abbreviations in this article. If possible, it is recommended to add an abbreviation summary.
Response 5: Thank you for your suggestion. The word template does not have an abbreviation section, but we summarized all abbreviations (list below) and will enquire about including the summary as part of the manuscript.
